# Feasibility and acceptability of the Indian Autism Screening Questionnaire in clinical and community settings

Nitin Antony[1], Aratrika Roy[1], Satabdi Chakraborty[2], Aparajita Balsavar[3], Amrita Sahay[4], Jaspreet S. Brar[5], Satish Iyengar[6], Triptish Bhatia[7], Vishwajit L. Nimgaonkar[8], Smita Neelkanth Deshpande[9]*

1 ICMR Project 'Development and Validation of the Screening Version of Indian Scale for Assessment of Autism', Centre of Excellence in Mental Health, Dept. of Psychiatric Social Work, ABVIMS-Dr. RML. Hospital, New Delhi, India, 2 Dept. of Psychiatric Social Work, Institute of Human Behaviour and Allied Sciences, New Delhi, India, 3 Bidar Institute of Medical Sciences, Rajiv Gandhi University of Health Sciences, Bengaluru, Karnataka, India, 4 National Institute for the Empowerment of Persons with Intellectual Disabilities (NIEPID), Noida, UP, India, 5 Department of Psychiatry and Consultant, Community Care Behavioural Health Organization, Western Psychiatric Hospital of UPMC, Pittsburgh, Pennsylvania, United States of America, 6 Department of Statistics, University of Pittsburgh, Pittsburgh, Pennsylvania, United States of America, 7 Indo-US Projects and NCU-ICMR, Centre of Excellence in Mental Health, Department of Psychiatry and De-addiction, ABVIMS-Dr. RML Hospital, New Delhi, India, 8 Department of Psychiatry and Department of Human Genetics, University of Pittsburgh School of Medicine and Graduate School of Public Health, Pittsburgh, Pennsylvania, United States of America, 9 Centre of Excellence in Mental Health, Department of Psychiatry, ABVIMS-Dr. RML Hospital, New Delhi, India

* smitadeshp@gmail.com

**Data Availability Statement:** All relevant data are within the paper and its Supporting Information files.

## Abstract

We developed and tested the Indian Autism Screening Questionnaire (IASQ), which was reported to be reliable and valid as compared to the Indian Scale for Assessment of Autism (ISAA) and the Childhood Autism Rating Scale -2 (CARS2). The present study describes the feasibility, acceptability, sociodemographic and developmental details of IASQ study participants in 5 settings- a psychiatry outpatients' clinic (n = 145), a specialised paediatric clinic (n = 24), a speciality disability centre (n = 174), a primary school (n = 41) and a government housing colony (n = 255). The IASQ could be easily administered and understood. Consistent with prior reports, the male-female ratio of participants with autism was 3.8:1. Developmental complications were reported more frequently in clinical settings, while delivery by Caesarean section was commoner among community-dwelling higher socioeconomic status mothers (53% of the officers' sample). Mothers of participants with autism more frequently reported Caesarean section birth for the proband ($\chi^2$ = 41.61, p < .0001) and prenatal and postnatal complications. Binary logistic regression confirmed that perinatal complications in the mother and father's (older) age at birth of the participant were associated with autism. The IASQ is a reliable, practical tool for screening for autism in clinical and non-clinical settings in India.

**Funding:** This work is supported by the Indian Council Medical Research (ICMR) Indian Council of Medical Research file number 5/4-4/151/M/2017/NCD-Dr Satabdi Chakraborty under Capacity Building Projects for National Mental Health Programme (ICMR-NMHP) Task Force and Fogarty International Center D43 TW009114 to Dr. Prof. Smita Neelkanth Deshpande. The funders had no role in study design, data collection and analysis, decision to publish, or preparation of the manuscript.

**Competing interests:** The authors have declared that no competing interests exist.

## Introduction

Community surveys for autism using a brief easy to administer tool followed by specialist evaluation and early intervention ensure better outcome in terms of intellectual and social adaptive functioning [1–3]. According to Boateng and colleagues [4] scale development needs to follow certain specific steps, of which the fourth step is administration and gathering sufficient data from the 'right people'. The 'right people' should include a range of potential target populations. They recommend at least ten respondents per survey item, and/or 200–300 observations in order to obtain sufficient data. Screening instruments need to be administered in different populations not only for pilot-testing and validation, but also to check for administration in terms of time required, ease of comprehension, feasibility and acceptability.

The results of prevalence surveys in India are presented in Table 1.

Government of India has a program 'DISHA' for children upto 10 years with disabilities covered under the National Trust Act and autism in included in it. It is an early intervention and school readiness scheme for children with disabilities. Severity of autism is determined by INCLEN followed by CARS2. Rajiv Bal Yojana has also included autism and INCLEN tool is used for autism.

To our knowledge, two Indian community-tested screening tools, validated against other acknowledged measures of autism have been developed. The Chandigarh Autism Screening Instrument (CASI- 2018- for children 1.5–10 years old) is a community-based 37-item Hindi screening tool with binary 'Yes/No' responses that takes around 15–20 minutes to administer. A shorter version titled, 'CASI Bref' consisting of 4 Yes/No items is also available [9]. This scale has adequate psychometric properties and is time- and cost-effective. The Trivandrum Autism Behaviour Checklist (TABC) was validated against the Child Autism Rating Scale (CARS 20-item scale each rated 1 to 4) for screening children between ages of 24–36 months [10].

Large Indian community surveys for autism have described prevalence, characteristic features, factors influencing course and perspectives of the autism community (autistic individuals, their care-givers, people coming in close contact and professionals) [7, 11]. Since autism is relatively less prevalent (about 2.25 per 1000 in one recent Indian urban study–Arun and Chavan 2021), it is cost-effective to combine other objectives with surveys. Thus while studying feasibility of Modified Checklist for Autism in Toddlers (MCHAT) the autism prevalence among Malaysian children aged 18 to 36 months was reported as 1.6 in 1000 [12], The same study also reported that mean age of fathers and mothers at the time of conception was 33.6 ±5.09 years and 31.6±4.99 years respectively. A Kerala case-control study of autism reported upper and upper middle socioeconomic status and male gender as risk factors, whereas rural place of residence was a protective factor [13]. However, there are widely differing estimates of autism prevalence probably depending on the awareness of the community as regards childhood neurodevelopmental disorders, their help-seeking behaviour, availability and adequacy of services and various other socio-demographic factors [14]. Testing a screening tool in different populations away from the hospital settings, is one way to circumvent these factors. The psychometric properties of the tool must be kept in mind as well. Selecting different populations can increase the generalizability of the tool. An appropriate screening tool is one that helps primary health care workers identify at-risk individuals in the community after due validation among participants of younger ages as well as adults [15].

The Indian Autism Screening Questionnaire (IASQ) is free-to-use, simple to administer and train [16]. The IASQ is a 10-item screening questionnaire so no domains are required as in ISAA, however, the first seven questions are from Social Relationship and Reciprocity domain of ISAA as these were the replied by largest number of participants in the original

**Table 1. Prevalence of autism in India.**

| Study and year | Study site | Sample size | Instrument used | Age group | Prevalence |
|---|---|---|---|---|---|
| Raina et al. 2017* [5] | Himachal Pradesh | 11000 | ISAA | 1–10 | 0.15% |
| Arora et al. 2018 [6] | Five States in north and west India | 3964 | INCLEN | 2–6 | 1 in 125 |
| | | | | 6–9 | 1 in 80 |
| | Overall in India | | | 2–9 | 1 in 89 |
| Poovathinal et al. 2016 [7] | Kerala | 5331 | Non-Standardized questionnaire | 1–10 | 23.3/10,000 |
| Deshmukh et al. 2013 [6] | South India, semiurban | | | 2–9 | 0.8–1.3% |
| Nair et al. 2014 [8] | Kerala | **101,438** | 0–6 | 0–6 | 12.8/1000 |

*the ISAA is not a screening instrument but a diagnostic and evaluative one.

ISAA study. The eighth and ninth questions belong to behaviour patterns domain and tenth from sensory aspects. These questions are noticed and complained by the parents most.

It has been validated against the Indian Scale for Assessment of Autism (ISAA) [15] and the Childhood Autism Rating Scale Version 2 (CARS2) [17] and found to have satisfactory psychometric properties. It is designed for use by primary health care workers with minimum training to enable screening of persons with probable autism in the community [15]. It was used in Delhi, India for validation by clinical psychologists and research personnel but can be used by primary health care workers after adopting it as screening tool in community.

The IASQ was initially developed and tested in a clinic population of 145 children and adolescents between the ages of 3–18 years (90 with autism and 55 with other psychiatric disorders), with male preponderance. A cutoff of 1 was determined as having the best sensitivity so as not to miss even one case from the general population.

However, since the IASQ was developed in a psychiatric setting it was important to test its feasibility, acceptability and reliability in other settings. The present paper describes its administration in five diverse samples. We examine its feasibility for administration in a community setting and in clinics and its acceptability as evaluated by respondents. Further, we examine the clinical and demographic factors among individuals identified as positive and negative for autism using the IASQ, as an indirect evaluation of its validity.

## Methodology

This paper is a part of a larger project which developed and explored the psychometric properties of the IASQ [15–18].

### Ethics

Ethical permission was obtained from Institutional Ethics Committee of this institution before the beginning of the study and renewed as required. Written informed consent and assent were obtained from participants and parents as appropriate.

Institutional Ethics Committee, ABVIMS, Dr. R.M.L. Hospital approved the study (letter no.191 (10/2017) /IEC/ ABVIMS/ RMLH)/92 dated 5th March, 2020). Permissions from school authorities were taken in case of school participants and permission from the Head of the Paediatrics Department and the specialist referring participants from Neurodevelopmental Clinic (NDC) was obtained in case of neurodevelopment participants. Permission from the Officer In charge of the National Institute for Empowerment of Persons with Intellectual Disability (NIEPID) was obtained to recruit non-psychiatric patients from that institution. In the community, announcements along with photos of research personnel and request for

participation were posted in the colony WhatsApp groups before initiating the survey and repeated from time to time. Research fellows always carried their ID cards. Procedure for school participation is described above.

## Training

All research personnel, students and interns administering the scale received training consisting of how to obtain informed consent and assent, and education about autism. Trainees also, viewed an hour-long training video of ISAA items (on which the IASQ is based), to familiarize themselves with individual IASQ items and its supplementary/ explanatory questions meant to aid the data collection process, and a undertook a mock IASQ administration. The training video can be accessed at https://www.youtube.com/watch?v=kz-rddmCEQQ&t=25s. Training for the IASQ can be both in person or online, lasting one day, but trainees need to view the video mentioned above on the YouTube. For the purpose of this study, we trained four research personnel, about 24 students and 10 interns at RMLH, while all teachers at NIEPID were trained. All were educated upto at least Masters in Psychology or MBBS (in case of interns).

## Settings for administration of the IASQ

This paper describes results of the IASQ from five diverse settings: two specialized hospital settings (a Psychiatry outpatients and a Paediatrics Neurodevelopmental clinic from a teaching hospital), a primary school, a specialized disability institution and a diverse community.

Participants were recruited for the study from a Psychiatry outpatient setting for the initial development of the screen followed by two non-psychiatric clinical settings and three general community settings. The first clinical group was recruited from the Psychiatry outpatient department (OPD) of a large tertiary care government teaching institution. The children with autism are referred for disability certificates or diagnosis. This sample was used to evaluate the psychometric properties of the IASQ [15]. The parents of the children with autism reporting for disability certification, intervention or diagnosis at the Department of Psychiatry, CEIMS, ABVIMS, Dr. R.M.L. Hospital were approached and explained the study. They were informed about the study and requested for participation by the treating clinician. If they agreed, they were directed to the research room where they were informed in detail about the study and written informed consent was obtained by the research personnel if they agreed to participate. Written assent was obtained from the children who were able to assent. Participants were recruited from two other non-psychiatric clinical settings which persons with autism were likely to attend: a once-weekly Neurodevelopmental Clinic (NDC) of the same institution in the department of Paediatrics, and a clinic at the National Institute for Empowerment of Persons with Intellectual Disability (NIEPID), caring for children with disability for assessment and training, in the Delhi region. After due permission from the Officer-in-Charge of the institute, the clinical psychologists at NIEPID were trained to administer the IASQ to all children and adolescents of appropriate ages presenting to their outpatient clinic [16]. They then referred those scoring above cut-off to the project staff for administration of the ISAA and/or CARS 2.

For the school survey, permission was obtained from the Principal of a Delhi primary-level school, after which parents were contacted, and their written informed consent was obtained. For those children whose parents consented, assent from the child/adolescent was then obtained. Research personnel also attended parent-teacher meetings to recruit and evaluate participants.

For further community participation, we conducted a house-to-house survey in the Chanakya Puri Government Officers' Colony (CGOC - each unit comprising an Officer's Housing

Unit (OHU) and the attached Servant Housing Unit (SHU). As per Central Public Works Department (CPWD) who are responsible for the maintenance of this colony, there are 376 OHUs and an equal number of attached SHUs. The housing units are double-storey houses for senior officers of Central Government of India. Officers of Indian Administrative Service, Indian Foreign Services, senior central government doctors, senior officers of the Indian Armed Forces occupy the OHU while the attached SHU is occupied by their domestic helpers. Thus, the OHU and SHU house two socioeconomically disparate groups who share some common community spaces and facilities. The survey was carried out at least three days per week, with time adjusted according to the convenience of householders.

Sample size for the community sample was calculated using the formula
Estimated Proportion/prevalence: - 0.01; Desired precision of estimate: - 0.01
Confidence level: - 0.90

**Formula:-** $\frac{Z^2 * P(1-P)}{e^2}$.

- $Z$ = *value from standard normal distribution for desired confidence level*

- *P is expected true proportion*

- *e is desired precision (half desired CI width).*

Using the above formula, our sample size was estimated to be 268. Assuming an attrition rate of 20%, we planned to screen 330 families of community dwellers conducting a door-to-door survey, including 50% in each stratum; OHU and SHU. We acquired the list of households from CPWD authority and used a two-stage sampling design to avoid selection bias and obtain best representation of population. In the first stage we used stratified sampling to divide households into different strata. Within each stratum, we placed similar types of households. In the second stage, to minimize selection bias we used random sampling to select 200 houses and visited each of them. If the selected unit owner refused, we approached the next number. If the OHU occupant agreed but SHU refused or *vice versa*, research staff approached the next similar housing unit. The IASQ was administered by one trained MBBS student and/or one research fellow.

From the other settings, all consenting parents and assenting children/adolescents were included.

## Statistical analyses

Frequencies and percentages were used to portray the total as well as group samples. Different participating groups were compared on sociodemographic details and developmental history using ANOVA for continuous variables and $\chi^2$ for categorical variables. Backward binary logistic regression was used to identify significant risk factors for autism among these sociodemographic and developmental variables. The persons with and without autism were also compared using t-tests and $\chi^2$ tests both in the total sample as well as in groups where adequate numbers were available.

## Sample

The sample comprised parents/caregivers who were willing to participate and provided written informed consent with assent from eligible children/adolescents.

## Procedure

Recruitment was carried out between September 2018 and March 2020 and from October 2020 to July 2021, the break being due to the COVID-19 pandemic. The procedure across all the settings was similar. In the Psychiatry outpatients' clinic, eligible parents were informed about the study by their treating physician. The IASQ was administered by a registered clinical psychologist. In the neurodevelopmental clinic, postgraduate Psychiatric Social Work interns or MPhil Psychiatric Social Work students administered the IASQ. In the school, IASQ was administered by 2 junior research fellows (general psychologists or social workers) recruited for the project.

In the community, announcements along with photos of research personnel and request for participation were posted in their WhatsApp groups before initiating the survey and repeated from time to time. Research fellows always carried their ID cards. Procedure for school participation is described above.

Basic socio-demographics and developmental history details were obtained followed by the administration of the IASQ to the parent. Some participants were additionally interviewed on the Indian Scale for Assessment of Autism (ISAA) and/or the Childhood Autism Rating Scale version 2 (CARS-2); these results are described elsewhere [15, 18].

## Tools used

1. Socio-demographic data sheet comprised basic identification details (based on ISAA manual), Identification numbers (ID), age and sex of the person with autism and their parents, years of education of both parents and person with autism, parental occupations, mother tongue, language spoken at home, family history of any mental illness, consanguinity and age of onset.

2. Developmental milestones information sheet (based on ISAA manual) elicited pregnancy details (nature of delivery, pre/peri/post-natal complications) and developmental history (age of attainment of motor and speech milestones), presence of regression and birth order.

3. Indian Autism Screening Questionnaire (IASQ) [15, 19] is a 10-item simple, easy-to-use screening tool to identify possible cases in the community. It was developed as a part of the ICMR Capacity Building Task Force and is designed for use for individuals aged 3 to 18 years. It is scored after interaction with the caregiver with or without observation of the person in question. Responses are recorded as yes or no. It takes approximately 10 minutes to administer. If more than one question is answered as 'yes' then the person is referred to a specialized centre for detailed evaluation by an expert.

**Data collection and management.** Participants were recruited, and data was collected from November 2018 to August 2021. Data was collected on paper scales described above. The identifiers like names etc. were not written on the questionnaires. These were entered in a different file and an ID was assigned to each participant. The data then was entered to iMann database. The database is password protected and only PIs and authorized research personnel can download or see the data. It is permanently stored at ICMR's Bio-Informatics Division's servers.

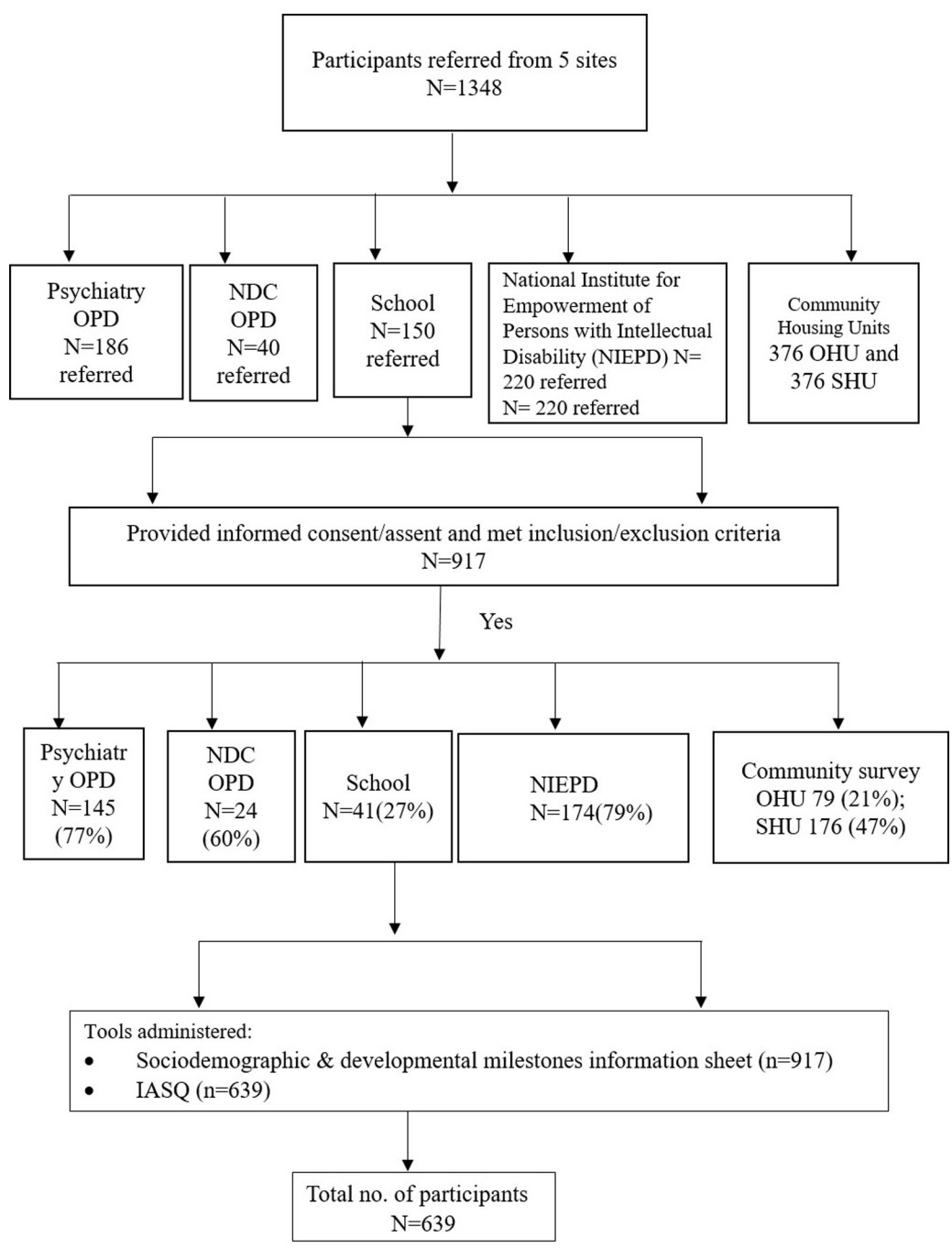

**Fig 1. Flowchart showing the recruitment process from all settings.**

## Results

### Sociodemographic

**Psychiatry outpatients' clinic.**  Out of a total of 186 referred, 145 participants consented to take part (78%) (Fig 1). There were 31 (21%) girls and 114 (79%) boys [15]. A total of 111 (77%) participants screened positive on IASQ, who were subsequently confirmed through ISAA.

**Table 2. Comparison of sociodemographic details of the sample across different settings (N = 917).**

| Sociodemographic factor | Psychiatry Out Patients (n = 145) | Neuro-developmental clinic (n = 24) | NIEPD (n = 174) | Community (Primary school) (n = 41) | Community (SHU n = 395) | Community OHU (n = 138) | F/χ2 value (p value) |
|---|---|---|---|---|---|---|---|
| Gender Male/Female | 114(79%)/31(21%) | 21(87%)/3(13%) | 104(60%)/70 (40%) | 30(73%)/11(27%) | 215(54%)/180 (46%) | 83(60%)/55(40%) | 35.58 (0.0001) |
| Screen autism negative/ screen autism positive | 34/111 | 7/17 | 64/110 | 40/1 | 344/3 | 814/3 | 442.63 (0.0001) |
| Average age of identified participants | 11.14(5.041) | 7.75(2.817) | 10.02(3.85) | 10.10(2.234) | 13.38 (5.97) | 17.57 (7.93) | 87.53 (0.0001) |
| Education of the participant in years | 2.94(3.895) | 0.38(1.245) | 2.87(3.676) | 2.76(1.513) | 6.77(4.22) | 10.70 (5.26) | 36.88 (0.0001) |
| Currently studying Yes/No | 118(81%)/27(19%) | 9(37%)/15(63%) | 155(90%)/18(10%) | 41(100%)/0 | 342(87%)/53 (14%) | 100(72%)/38 (28%) | 65.07 (0.0001) |
| Father's current age | 42.52(6.78) | 37.08(5.52) | 40.28(5.98) | 39.85(4.18) | 40.25 (6.52) | 51.10 (7.62) | 65.1 (0.0001) |
| Mother's current age | 38.85(6.70) | 33.08(6.05) | 36.90(8.43) | 36.48(4.34) | 35.89(5.84) | 47.79(7.34) | 70.30 (0.0001) |
| Father's years of education | 13.08(4.54) | 10.75(4.83) | 10.86(5.95) | 12.88(4.49) | 8.55(3.42) | 18.32(13.22) | 48.43 (0.0001) |
| Mother's years of education | 12.783(4.96) | 10.333(5.15) | 10.563(6.17) | 11.90(3.48) | 6.20(4.4) | 17.391(2.19) | 135.9 (0.0001) |

**Neurodevelopmental clinic.** In this extremely busy clinic, very few cases were referred by treating doctors. In addition, parents were often in a hurry to complete hospital procedures. Out of 40 referrals, only 24 consented to participate (Fig 1). Here too boys (n = 21) (87%)) outnumbered girls (n = 3) (13%). Out of 24,17 (42.5%) screened positive for autism (Table 2).

**National Institute for Empowerment of Persons with Intellectual Disability (NIE-PID).** Persons from the appropriate age groups, presenting for disability assessment, intervention and certification were referred by the staff psychologist to research fellows for consenting and evaluation. A total of 220 were referred and 174 (79%) consented for participation; out of which 104 (60%) were boys and 70 (40%) were girls (Fig 1 and Table 2).

**Primary school.** A primary school was approached and permission to administer IASQ was obtained from authorities. The consent forms with details were sent to the homes of children and those who agreed to participate were contacted on parent-teacher meeting day for all research procedures. Consent forms were sent to 150 parents but only 41(27%) participated in the study: boys 29 (72%) and girls 11 (28%). Only one child screened positive for autism (Fig 1).

**Government colony.** Based on random numbers we were to approach 400 (200 each of OHU and SHU) but since many persons refused or were not available; we approached all 752 units (376 each). Out of these a total of 255–79 (21%) OHU and 176 (47%) SHU families consented. Offspring of all ages of the family were included while asking about sociodemographic and autism related data.

A total of 533 offspring participants from these community families whose parents provided required details were included. Their data was obtained on the research instruments but 102 families did not continue after providing sociodemographic data. Data for a total of 431/533 offspring (81%) was obtained, and IASQ administered to the parents. All but three were screen negative on the IASQ.

The demographic details and developmental history of the participants of OHU and SHU are presented separately (Figs 1 and 2). Fathers and mothers residing in the OHU–as expected-were highly educated while those of SHU were minimally educated.

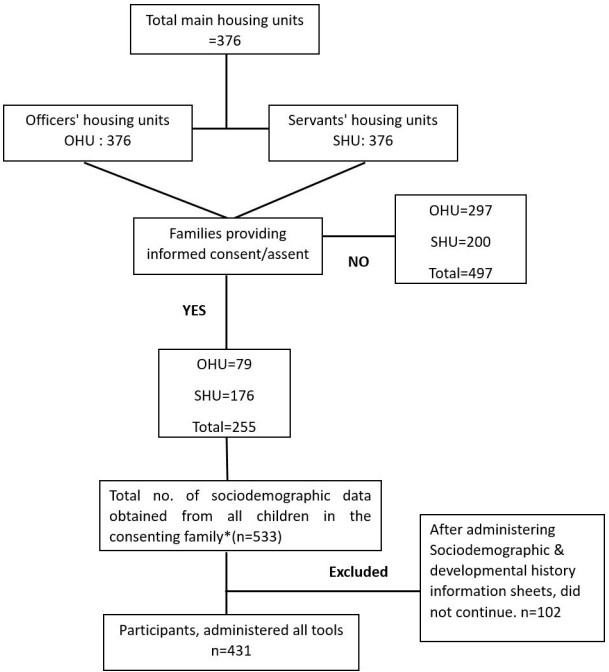

**Fig 2. Flowchart showing the recruitment process from community.**

In all the groups, males significantly outnumbered females ($\chi^2$ (df 5, N = 916) = 35.58, p<0.0001). The participants from neurodevelopmental clinic were the youngest and OHU were the oldest.

Pregnancy, delivery and pre and post-natal history of all samples were compared. Though maximum caesarean cases were reported in the OHU (53%) community sample, developmental complications were reported more frequently in the Psychiatry outpatients and NIEPD participants. There were no significant differences on neck holding, sitting, crawling and walking milestones among the different samples (recall by parent) (Table 3).

## Sociodemographic and developmental factors of participants with and without autism

We compared the participants' sociodemographic and developmental history on the basis of whether they were screened positive/negative on IASQ at cut-off point 1. As expected, the largest number of participants diagnosed with autism were from the clinics, while participants not diagnosed with autism were recruited from the community (Table 4).

**Logistic regression with outcome variable as autism present/absent (Total sample).**
We conducted backward variable selection logistic regression to study the sociodemographic and developmental risk factors for autism in our total sample. The outcome variable was taken as autism present/absent based on an IASQ cut-off of one. The predictor variables selected were gender of the child, age of father and mother, age of the parents at the time of delivery, education of parents, family history present/absent, consanguinity present/absent, delivery natural/caesarean, pregnancy eventful/uneventful, prenatal, natal or postnatal complications present or absent. Holding all other predictor variables constant, the odds of autism in males compared to females was 2.18 (95% CI [1.43,.3.34]).

**Table 3. Comparison of developmental history of the participants from different settings (N = 533).**

| Variables | Dept of Psychiatry (n = 145) | Neuro-developmental clinic (n = 24) | NIEPD (n = 174) | Community (Primary school, n = 41) | Community (SHU n = 395) | Community OHU (n = 138) | F/ $\chi^2$ value (p value) |
|---|---|---|---|---|---|---|---|
| Type of Pregnancy Eventful/uneventful | 22(15%)/123 (85%) | 1(4%)/23(96%) | 24(14%)/150 (86%) | 1(2%)/40(98%) | 42(11%)/353(89%) | 6(4%)/132(99%) | 13.59(0.02) |
| Nature of birth Normal/ Caesarean | 91(64%)/52(36%) | 18(75%)/6(25%%) | 115(68%)/54 (32%) | 26(63%)/15(37%) | 368(93%)/27(7%) | 64(47%)/73(53%) | 145.21 (0.0001) |
| Pre-natal Complications Absent/present | 119(83%)/24 (17%) | 23(96%)/1(4%) | 147(87%)/22 (13%) | 39(95%0)/2(5%) | 386(98%)/9(2%) | 125(91%)/12(9%) | 40.75 (0.0001) |
| Natal Complications Absent/Present | 120(84%)/23 (16%) | 22(92%)/2(8%) | 145(87%)/22 (13%) | 39(95%0)/2(5%) | 387(98%)/8(2%) | 129(94%)/8(6%) | 42.58 (0.0001) |
| Post-natal Complications Absent/ Present | 82(58%)/60(42%) | 15(62%)/9(37%) | 101(60%)/67 (40%) | 37(90%)/4(8%) | 381(96%)/14(4%) | 119(87%)/18(13%) | 169.72 (0.0001) |

Holding all other predictor variables constant, the odds of autism present in participants without family history of autism as compared those with positive family history of autism was 0.31 (95% CI [0.17, 0.56]. Similarly, absence of pre-natal and post-natal complications predicted a higher possibility of autism (OR.0.47(95%CI 0.25,0.89); (OR0.31 (95% CI 0.13, 0.33). Holding all other predictors constant, father's current age and father's higher age at child birth were associated with autism (OR0.89(95%CI 0.86,0.94) (OR 1.12 (95% CI 1.06, 1.18). The mother's higher education was also significant predictor variable for autism (OR 1.14(95%1.1, 1.18) in this study The logistic regression results confirm the univariate results presented.

**Demographics of participants.** Participants with autism were younger (F (1,813) = 55.56, p<0.0001), more males ($\chi^2$ (df 1, N = 814) = 32.34, p<0.0001), less educated (F (1,813) = 201.02, p<0.0001), and fewer currently studying ($\chi^2$ (df 1, N = 814) = 61.1, p<0.0001). While 95% of non-autism participants were currently studying, only 78% of participants with autism

**Table 4. Comparison of participants with autism and without autism (on IASQ) on sociodemographic and developmental factors (Total sample).**

| | IASQ negative for autism (n = 569) | IASQ positive for Autism(n = 245) | F/$\chi^2$ value | df | p value |
|---|---|---|---|---|---|
| Age of the participant | 12.20(4.13) | 9.75(4.70) | 55.56 | 1,813 | <0.0001 |
| Gender Male/Female | 314(55%)/255(45%) | 187(76%)/58(24%) | 32.34 | 1 | <0.0001 |
| Education of the participant | 5.98(3.9) | 1.92(3.38) | 201.02 | 1,813 | <0.0001 |
| Participant currently studying Yes/No | 543(95%)/26(5%) | 190(78%)/55(22%) | 61.1 | 1,813 | <0.0001 |
| Father's current age | 40.84(6.26) | 40.77(6.73) | 0.02 | 1,751 | 0.887(NS) |
| Mother's current age | 36.67(5.85) | 37.74(8.68) | 3.74 | 1,756 | 0.054(NS) |
| Father's age at birth of child | 28.7(5.59) | 31.07(5.76) | 26.41 | 1,751 | <0.0001 |
| Mother's age at birth of child | 24.45(5.33) | 28.05(8.18) | 49.97 | 1,756 | <0.0001 |
| Father's years of education | 10.12(4.82) | 13.23(4.72) | 71.97 | 1,696 | <0.0001 |
| Mother's years of education | 8.65(5.72) | 12.89(5.11) | 100.42 | 1,364 | <0.0001 |
| Consanguinity Present/absent | 8(1%)/558(99%) | 5(2%)/237(98%) | 0.456 | 1 | 0.545(NS) |
| Family History present/absent | 37(7%)/529(93%) | 51(21%)/191(79%) | 36.91 | 1 | <0.0001 |
| Type of Pregnancy Eventful/uneventful | 64(11%)/498(89%) | 26(11%)/217(89%) | 0.81 | 1 | 0.81(NS) |
| Nature of delivery Normal/Caesarean | 462(82%)/102(18%) | 148(61%)/96(39%) | 41.61 | 1 | <0.0001 |
| Pre-natal Complications Absent/Present | 538(95%)/26(5%) | 203(83%)/41(17%) | 33.29 | 1 | <0.0001 |
| Natal Complications Absent/Present | 538(95%)/26(5%) | 207(86%)/35(14%) | 23.5 | 1 | <0.0001 |
| Post-natal Complications Absent/Present | 498(88%)/66(12%) | 140(58%)/102(42%) | 95.14 | 1 | <0.0001 |

were doing so. This was true for individual groups analysed separately as well (S1 and S2 Tables).

**Parental factors.** There was no difference on parents' current age in the autism/no autism groups. Parents of participants without autism were more educated. No significant difference on fathers' current age was noted between no autism and autism groups among the psychiatry OPD group while mothers of participants with autism were older than those of no autism (F $(1,141) = 4.33$; p 0.04). Both father's and mother's age at the time of birth of their offspring was significantly different between the two groups in the total sample (fathers' ages- autism/ non-autism: F $(1,751) = 26.41$, p<0.0001) (mothers' ages autism/non-autism: F $(1,756) = 49.97$, p<0.0001). This differed among samples. While fathers' age at birth of the offspring was not significantly different in the samples from Psychiatry OPD and NIEPID, mothers' age at birth was significantly different between autism/non-autism groups in both the psychiatry OPD sample (F $= 5.39$ $(1, 95)$; p $= 0.022$) as well as the NIEPID samples (F$(1,150) = 5.92$; p $= 0.016$).

Both parents of the autism group were more educated than those of the group not diagnosed with autism (p $< .0001$). With regard to NDC participants, there was no significant difference between autism and no autism on any sociodemographic variable. There were 5 males and 2 females in autism group and 16 males and one female in the group identified as no-autism in the NDC sample. Though males were proportionately more in the autism group, no statistical analysis was performed due to the small sample from the NDC.

**Pregnancy and perinatal factors.** There were no significant differences with regard to consanguinity and type of pregnancy among the autism/no autism groups in the total sample. Analysed separately (only Psychiatry OPD and NIEPD samples were analysed) (no significant consanguinity difference) while other three samples were too small for statistical analyses on this variable. There were only three participants with autism in OHU and SHU samples, hence no statistical analyses were done.

The type of pregnancy was not significantly different in Psychiatry OPD sample while in NIEPID sample, pregnancy was significantly more eventful in case of participants with autism. Other groups were not analysed due to their small size.

A greater number of autism cases reported family history of the disorder ($\chi^2. = 36.91$ (df $= 1$, N $= 808$), p<0.0001) in the total sample only. In all subsamples, family history was not a significantly differing variable. Mothers reported significantly higher number of caesarean deliveries among participants with autism both in total sample ($\chi^2 = 41.61$ (df $= 1$, N $= 808$), p<0.0001) and psychiatry clinic sample ($\chi^2 = 7.94$ (df $= 1$, N $= 169$), p $= 0.006$) and NIPIED sample ($\chi^{2.} = 7.662$ (df $= 1$, N $= 143$), p $= 0.006$). Mothers of autism participants reported greater complications during prenatal ($\chi^{2.}$ (df $= 1$, N $= 808$), $= 33.29$,p<0.001), natal ($\chi^{2.} = 23.5$ (df $= 1$, N $= 806$), p<0.001) and postnatal ($\chi^{2.} = 95.14$ (df $= 1$, N $= 806$), p<0.001) period when analysed in the total sample only (Table 4).

**IASQ training and administration.** All IASQ administrators including interns and students, could grasp the concept of autism and learnt to administer the IASQ with brief training. Average administration time for all instruments combined was approximately 10–15 minutes. The IASQ was easy to administer across all settings.

**Parental understanding of terms describing symptoms of autism.** Parents at the two hospital-based clinics had no difficulty in comprehending IASQ questions. In the community survey, both officers, their families and domestic helpers and families found the questions comprehensible. Thus, the vast majority of parents could understand the terms and answer accurately. However, at the primary school parent-teacher interviews, a few parents were unable to comprehend the meaning of IASQ questions 2, 3 and 7 concerning 'lacking social smile', 'remains aloof', and 'unable to initiate or sustain conversation with others'. Strangely, at the rehabilitation centre for persons with Intellectual Disability also, several parents were

unaware of these particular symptoms. The symptoms were described further, using a set of indicative questions we had developed beforehand (Chakraborty et al., 2021). A few parents confused terms like 'aloofness', 'inability or difficulty in initiating or maintaining conversations', 'lack of social smile', and 'inappropriate eye contact' with typical adolescent developmental behaviour. Parents misconstrued the question 'does your child remain aloof', interpreting the aloofness as the somewhat indifferent, socially withdrawn behaviour that some adolescents may demonstrate.

## Discussion

We investigated the feasibility, acceptability and reliability of the Indian Autism Screening Questionnaire in various samples. Persons are more frequently referred to the Psychiatry out-patients for evaluation, diagnosis and disability certification of behavioural disorders. Hence the largest number of diagnosed participants were from this setting followed by the other two clinical settings. In the government colony however, we could recruit only three participants with autism. Recruitment from the primary school was low. In spite of the current integrated education policy, young children with autism do not attend any school or are sent to special schools. This probably accounted for both the low recruitment in the school as well as lack of knowledge about autism symptoms by the parents.

Experts in scale development recommend that potential scale items should be tested on a heterogeneous large sample starting with clinical sample first and then the general population [4, 20].

### Demographics of participants

We examined known risk factors for autism in all the settings. Males predominated in all settings. Literature has reported male-to-female ratio as 4:1 among persons with autism [21–23]; one study reported male to female ratio of 4.7:1 [24]. Our study reported 3.8:1 male-to-female ratio, which is similar. Fewer participants with autism were currently studying suggesting some either had very severe symptoms, or were too young or studied in special schools.

### Demographics of parents

The community participants (OHU) and their parents were most educated as they were higher ranking government officers. The second community category consisted of domestic helpers who were less educated. Our study also supports the role of family history in autism [25]. There were significantly more participants with family history of autism in the autism group than non-autism group although overall, there were fewer participants with a positive family history. Others have observed that family history of other neurological disorders is also higher in persons with autism [25].

Higher ages of father and mother were associated with greater chance of autism among our participants. However, after logistic regression only father's higher age remained significant. Other studies reported that mother's higher age is associated with 41% increased risk of autism while father's higher age is associated with 55% greater risk of autism [26].

There are reports that parental consanguinity can be a risk factor for autism but some studies have not found this association [27, 28]. Our study did not find any association of consanguinity with autism in our sample both in univariate analyses and logistic regression.

## Perinatal factors

Prenatal and postnatal complications were reportedly higher in our clinical samples than among our community samples. Among community samples pre-natal, natal and post-natal complications were reported more frequently in SHU than in OHU families. Studies suggest that low socioeconomic status results in complications at various stages of pregnancy and delivery [29]. This may be due to prolonged working hours, costs of transportation to reach hospital for delivery and expensive medical care, seeking inappropriate pre- and post-natal care due to less education.

We observed that prenatal, natal and perinatal complications were reported more frequently in mothers of autistic participants than in mothers of non-autistic participants. These factors were confirmed by logistic regression also. Our findings are supported by several other studies. A meta-analysis in which 17 studies were included with data from 37,634 autistic participants and 12,081,416 non-autistic participants confirmed the relationship between some prenatal, perinatal, and postnatal factors with autism. However, it was not clear whether these are causal factors or secondary [30]. Another study confirmed these results [31]. Genetic studies reported that about 35–40% autism is caused by genetic factors and the rest may be other factors such as prenatal, perinatal and postnatal environmental factors [32–35].

The largest number of caesarean deliveries were in the OHU community group. This may be due to the societal trend in higher socioeconomic class. Older women, higher educated mothers, residing in urban areas and, belonging to high socio-economic status are said to tend to opt for voluntary C-section deliveries and seek private institutional delivery [36]. In India, a study reported that caesarean rates move from 4.4% to 35.9% as women move up the social ladder [37]. This may be because women from higher socioeconomic groups go to private institutions for deliveries and they have higher rates of caesareans [38].

Feasibility is important for a test or tool when there are multiple tests or tools available that are valid, reliable, time and cost effective. The IASQ was developed from the ISAA which has good reliability, validity and feasibility. According to Centers for Disease Control and Prevention USA (CDC), there are some myths regarding screening for autism. Some of these are – 'a great deal of training is needed to administer screening correctly'; 'screening takes a lot of time' and 'tools that incorporate information from the parents are not valid'. Contrary to this view, screening by IASQ does not require much training, takes only 10–15 minutes and information from parents is reliable as they have observed their child over a long period of time [39].

In all, we found participants from Psychiatry OPD, NDC, and NIEPID were better informed and acquainted with the nature of the presenting problems of their offspring. The schools we approached probably had few or no students with similar developmental disabilities, because such children are preferentially sent to special schools. This difference in the level of awareness may account for the challenges faced in the acceptability and ease of administration in the latter settings and may hinder parents from availing of specialist consultations [40].

Bauer et al. evaluated assessment tools for autism for their feasibility in LMICs and reported that most of these tools are not suitable for use in LMIC due to cost, copyright issues, restrictions on who can purchase and administer these tools and readability levels. In addition, these instruments do not have local norms. Furthermore, the assessments reviewed do not have local norms either. Only the M-CHAT-R/F can be used but it has narrow age range [41]. The IASQ is free to use and very short. It can be easily translated in the local language and adapted for use locally.

The feasibility of the scale was evident by its ease of administration in multiple settings in varied situations. In the community survey, despite constraints of space and time the scale was

well received by the participants. They were able to comprehend the questions and were able to complete the procedure without any delay.

We also compared two community groups who were socioeconomically very different though living in the same area. More of SHU families agreed to participate. This in accordance with the studies which indicate that health survey participation is higher in lower socio-economic groups as also among those with less education [42, 43]. The OHU group in our data was educated consisting of high rank government officials and though we tried to contact them at different times, their availability at home was hampered by their long duty hours and working hours of both spouses. In addition, the higher status persons were more suspicious and cautious of inviting strangers at home as fewer OHU participated than SHU. Moreover, the study was also constrained by the pandemic and lockdowns.

There were several potential limitations in our study. Data were collected from the retrospective memories of the parents for historical facts although they were quite clear about their child's symptoms. Many parents could not recall the prenatal, perinatal and postnatal events clearly but recall bias is in favour of perinatal events, being recalled at high frequencies for perinatal trauma-related outcomes. We selected only those parents who were available and consented to participate. We did not have any data for the parents who were not available or did not consent.

## Conclusion

IASQ was feasible for use in community as well as hospital settings as its questions were easy to comprehend. IASQ can be used to broaden the reach for autism screening and help in getting timely diagnosis and intervention. In addition, the community identification of persons with autism can help identify risk factors and prevalence of the condition. Our survey confirms the differences between clinical and non-clinical, and between high and low socioeconomic status populations in developmental factors. The IASQ can also be used to train and educate the public. There is great need to educate the public about autism as well as work towards removing the stigma surrounding mental illness and ensure inclusiveness and empowerment of individuals with disability. Community surveys help in collecting useful public health information.

## Supporting information

**S1 Table. Comparison of children with autism and without autism on sociodemographic and developmental factors in the Psychiatry OPD sample.**
(DOCX)

**S2 Table. Comparison of children with autism and without autism on sociodemographic and developmental factors NIEPD sample\*.**
(DOCX)

**S1 File.**
(SAV)

**S2 File.**
(SAV)

## Acknowledgments

This work is supported by the Indian Council Medical Research (ICMR) under Capacity Building Projects for National Mental Health Programme (ICMR-NMHP) Task Force. We

thank Dr Soumya Swaminathan (then Secretary, Dept. of Health Research, DHR), Dr Balram Bhargav (former Secretary DHR), Dr Ravinder Singh and Dr. Harpreet Singh, ICMR. We thank the faculty of 'Cross-Fertilized Research Training for New Investigators in India and Egypt' and Psychiatric Research for Intervention, Implementation in India (PRIIIA). We thank the National Coordinating Unit for logistic support and the ICMR Data Management Unit for designing the database.

## Author Contributions

**Conceptualization:** Triptish Bhatia, Smita Neelkanth Deshpande.

**Data curation:** Triptish Bhatia.

**Formal analysis:** Triptish Bhatia.

**Funding acquisition:** Satabdi Chakraborty.

**Investigation:** Nitin Antony, Aratrika Roy, Aparajita Balsavar, Amrita Sahay.

**Methodology:** Smita Neelkanth Deshpande.

**Project administration:** Triptish Bhatia.

**Supervision:** Satabdi Chakraborty, Triptish Bhatia.

**Validation:** Jaspreet S. Brar, Satish Iyengar, Vishwajit L. Nimgaonkar.

**Visualization:** Jaspreet S. Brar, Satish Iyengar, Vishwajit L. Nimgaonkar.

**Writing – original draft:** Nitin Antony.

**Writing – review & editing:** Jaspreet S. Brar, Satish Iyengar, Triptish Bhatia, Vishwajit L. Nimgaonkar, Smita Neelkanth Deshpande.

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
