## [Decision Letter · Decision Letter 0]

8 Aug 2023

PONE-D-22-30308Feasibility and acceptability of the Indian Autism Screening Questionnaire in clinical and community settingsPLOS ONE

Dear Dr. Deshpande,

Thank you for submitting your manuscript to PLOS ONE. After careful consideration, we feel that it has merit but does not fully meet PLOS ONE’s publication criteria as it currently stands. Therefore, we invite you to submit a revised version of the manuscript that addresses the points raised during the review process.

We look forward to receiving your revised manuscript.

Kind regards,

Engy Asem Ashaat

Academic Editor

PLOS ONE

Journal Requirements:

"This work is supported by the Indian Council Medical Research (ICMR) under Capacity Building Projects for National Mental Health Programme (ICMR-NMHP) Task Force. We thank Dr. Soumya Swaminathan (then Secretary, Dept. of Health Research, DHR), Dr. Balram Bhargav (former Secretary DHR), Dr Ravinder Singh and Dr. Harpreet Singh, ICMR. We thank the faculty of ‘Cross-Fertilized Research Training for New Investigators in India and Egypt’ (D43 TW009114, HMSC File No. Indo41 Foreign/35/M/2012-NCD-1, funded by Fogarty International Centre, NIH). We thank the National Coordinating Unit for logistic support and the ICMR Data Management Unit for designing the database. The content of this manuscript is solely the responsibility of the authors and does not necessarily represent the official views of NIH or ICMR. NIH and ICMR had no role in the design and conduct of the study; collection, management, analysis, and interpretation of the data; preparation, review, or approval of the manuscript; and decision to submit the manuscript for publication."

"This work is supported by the Indian Council Medical Research (ICMR) under Capacity Building Projects for National Mental Health Programme (ICMR-NMHP) Task Force."

Please include your amended statements within your cover letter; we will change the online submission form on your behalf."""

"This work is supported by the Indian Council Medical Research (ICMR) under Capacity Building Projects for National Mental Health Programme (ICMR-NMHP) Task Force."

Please respond by return e-mail so that we can amend your financial disclosure and competing interests on your behalf.

7. Please upload a copy of Supplementary Tables 1 & 2 which you refer to in your text on page 19. 

Reviewers' comments:

Reviewer's Responses to Questions

**Comments to the Author**

1. Is the manuscript technically sound, and do the data support the conclusions?

Reviewer #1: Yes

Reviewer #2: No

2. Has the statistical analysis been performed appropriately and rigorously? 

Reviewer #1: Yes

Reviewer #2: I Don't Know

3. Have the authors made all data underlying the findings in their manuscript fully available?

Reviewer #1: Yes

Reviewer #2: No

4. Is the manuscript presented in an intelligible fashion and written in standard English?

Reviewer #1: Yes

Reviewer #2: No

5. Review Comments to the Author

Reviewer #1: Comments

Introduction

-There is mention of Indian surveys in autism, it would add value if you can bring in figures from the Indian surveys on prevalence and incidence rate of autism in India.

-Would be great for a reader to understand the broad ten components of IASQ

-Line 107- It is used by primary health worker…..? in which country is it used? In which sector it is used by primary health workers? Are there any government programs or schemes which helps in early detection and identification of Autism in India. If any, please mention the tool administered by them?

- To increase the readability, it would be great to mention the five diverse samples briefly based on the setting you are going to test the tool

-Please brief on the research question: Importance of testing the tool beyond psychiatric setting as the authors have mentioned it as free, simple and easy to train tool in line 104

Methods

-Ethics permissions- please mention the name of IEC with date and reference number, may be crucial as children are involved. Also mention about the additional permissions required for the study

-How was the consent from psychiatric setting obtained

Training

-Was the training in person or online, and how many days and how many participants, period of training? Qualification of researches trained?

Setting

-Mention the period of recruitment and data collection.

-Please mention how was the data collected data managed and privacy was ensured

Results

Table 1- community primary school n=41 please check n. if n=40 then overall N has to be reduced to 916

Discussion

-Please bring in context of LMICs to the discussion part

Reviewer #2: The authors developed and tested the feasibility, acceptability, sociodemographic and developmental details of the Indian Autism Screening Questionnaire (IASQ). They reported as a reliable and valid scale compared with the Indian Scale for Assessment of Autism (ISAA) and the Childhood Autism Rating Scale-2 (CARS2).

The authors found that the IASQ to be easily administered and understood. They did they in different settings: psychiatric outpatients ‘clinic, specialized pediatric clinic, a disability center, a primary school and a government housing colony.

Not surprisingly they found that factors such as developmental, prenatal and postnatal complications were more associated with autism.

Comments;

While the authors did a large study on several setting, it is not clear HOW was the diagnosis of autism done. They applied a screening test, but that is not a diagnostic tool.

The major issue with this work is that the authors did not report how was the autism diagnosis actually made.

They must explain how the kids were who tested positive in the IASQ were finally diagnosed with autism. Did they follow the DSM-5 guidelines? Were ADOS/ADIR done to complement the diagnosis.

Also, their flowchart are very confusing with no legends, they should include the final patients who actually entered the study.

6. PLOS authors have the option to publish the peer review history of their article (what does this mean?). If published, this will include your full peer review and any attached files.

Reviewer #1: No

Reviewer #2: No

---

## [Author Response · Author response to Decision Letter 0]

19 Sep 2023

Comments 

Introduction

Query: There is mention of Indian surveys in autism, it would add value if you can bring in figures from the Indian surveys on prevalence and incidence rate of autism in India.

Reply: Thank you for this suggestion. We have made the table of studies and have added in the manuscript in introduction as follows:

The results of prevalence surveys in India are presented in Table1.

Table 1: Prevalence of Autism in India

Study and year Study site Sample size Instrument used Age group Prevalence

Raina et al. 2017(1)*

Himachal Pradesh 11000 ISAA 1-10 0.15% 

Arora et al.2018(2)

 Five States in (i) north and (ii) west India 

(iii) Overall in India 3964 INCLEN 2-6

6-9

2-9 1 in 125

1 in 80

1 in 89

Poovathinal et al.2016(3)

Kerala 5331 Non-Standardized questionnaire 1-10 23.3/10,000

Deshmukh Arora et al. 2013(2)

South India, semiurban 2-9 0.8-1.3%

Nair et al.2014(4)

Kerala 101,438 0-6 0-6 12.8/1000

*The ISAA is not a screening instrument but a diagnostic and evaluative one.

Query: Would be great for a reader to understand the broad ten components of IASQ. 

Reply: Thank you for this suggestion. We have added the following text in the description of the IASQ.

The IASQ is a 10-item screening questionnaire, so no domains are required as in ISAA, however, the first seven questions are from Social Relationship and Reciprocity domain of ISAA as these were the replied by largest number of participants in the original ISAA study. The eighth and ninth questions belong to behavior patterns domain and tenth from sensory aspects. These questions are noticed and complained by the parents the most.

Query: Line 107- It is used by primary health worker…..? in which country is it used? In which sector it is used by primary health workers? 

Reply: Apologies for the confusion. This paper describes the use of this new instrument IASQ in the community. The 107 line says, “It is designed for use by primary health care workers with minimum training to enable screening of persons with probable autism in the community [12].” We have now modified it as follows:

It was used in Delhi, India for validation by clinical psychologists and research personnel but can be used by primary health care workers after adopting it as screening tool in community.

Query: Are there any government programs or schemes which helps in early detection and identification of Autism in India. If any, please mention the tool administered by them? 

Reply: Yes, there are some programs. We have added the following in the introduction:

Government of India has a program ‘DISHA’ for children upto 10 years with disabilities covered under the National Trust Act and autism in included in it. It is an early intervention and school readiness scheme for children with disabilities. Severity of autism is determined by INCLEN followed by CARS2. Rajiv Bal Yojana has also included autism and INCLEN tool is used for autism.

Query: To increase the readability, it would be great to mention the five diverse samples briefly based on the setting you are going to test the tool. 

Reply: Thank you. We briefly mentioned these diverse settings and samples in our abstract. We begin our settings section now as follows: 

This paper describes results of the IASQ from five diverse settings: two specialized hospital settings (a Psychiatry outpatients and a Paediatrics Neurodevelopmental clinic from a teaching hospital), a primary school, a specialized disability institution and a diverse community. We have included different populations to increase the generalizability of the tool.

Query: Please brief on the research question: Importance of testing the tool beyond psychiatric setting as the authors have mentioned it as free, simple and easy to train tool in line 104 

Reply: respectfully to our understanding, we have addressed this query in our sentence describing Boateng’s steps in scale development. However, we have added the following text at the end of paragraph 1 in the introduction section:

However, there are widely differing estimates of autism prevalence probably depending on the awareness of the community as regards childhood neurodevelopmental disorders, their help-seeking behaviour, availability and adequacy of services and various other socio-demographic factors[14]. Testing a screening tool in different populations away from the hospital settings, is one way to circumvent these factors. The psychometric properties of the tool must be kept in mind as well. Selecting different populations can increase the generalizability of the tool. An appropriate screening tool is one that helps primary health care workers identify at-risk individuals in the community after due validation among participants of younger ages as well as adults.[15]

Methods 

Query: Ethics permissions- please mention the name of IEC with date and reference number, may be crucial as children are involved. Also mention about the additional permissions required for the study 

Reply: Thank you. We took every care for ethical conduct of this research. Apart from the parental consent and assent from the child to the extent possible) we obtained permission from the respective authorities as well, as added/described already, below:

Institutional Ethics Committee, ABVIMS, Dr. R.M.L. Hospital approved the study (letter no.191(10/2017)/IEC/ABVIMS/RMLH)/92 dated 5th March, 2020). Permissions from school authorities were taken in case of school participants and permission from the Head of the Paediatric Department and the specialist referring participants from Neurodevelopmental Clinic (NDC) was obtained in case of neurodevelopment participants. Permission from the Officer In charge of the National Institute for Empowerment of Persons with Intellectual Disability (NIEPID) was obtained to recruit non-psychiatric patients from that institution.

The following text (already present in the text) has been highlighted. 

In the community, announcements along with photos of research personnel and request for participation were posted in the colony WhatsApp groups before initiating the survey and repeated from time to time. Research fellows always carried their ID cards. Procedure for school participation is described above.

Query: How was the consent from psychiatric setting obtained? 

Reply: Thank you. We have expanded the description for this issue as follows: 

The first clinical group was recruited from the Psychiatry outpatient department (OPD) of a large tertiary care government teaching institution. The children with autism are referred for disability certificates or diagnosis. This sample was used to evaluate the psychometric properties of the IASQ [12]. The parents of the children with autism reporting for disability certification, intervention or diagnosis at the Department of Psychiatry, CEIMS, ABVIMS, Dr. R.M.L. Hospital were approached and explained the study. They were informed about the study and requested for participation by the treating clinician. If they agreed, they were directed to the research room where they were informed in detail about the study and written informed consent was obtained by the research personnel if they agreed to participate. Written assent was obtained from the children who were able to assent.

Training 

Query: Was the training in person or online, and how many days and how many participants, period of training? Qualification of researches trained?

Reply: Thank you for pointing at this gap. We have explained and modified the text as follows now: 

Training for the IASQ can be both in person or online, lasting one day, but trainees need to view the video mentioned above on the YouTube. For the purpose of this study, we trained four research personnel, about 24 students and 10 interns at RMLH, while all teachers at NIEPID were trained. All were educated upto at least Masters in Psychology or MBBS (in case of interns).

Setting

Query: Mention the period of recruitment and data collection.

Reply: Thank you. We have added and expended the text as follows:

Data Collection and Management: Participants were recruited, and data was collected from November 2018 to August 2021. Data was collected on paper scales described above. The identifiers like names etc were not written on the questionnaires. These were entered in a different file and an ID was assigned to each participant. The data then was entered to iMann database. The database is password protected and only PIs and authorized research personnel can download or see the data. It is permanently stored at ICMR’s Bio-Informatics Division’s servers.

Query: Please mention how was the data collected data managed and privacy was ensured 

Reply: Thank you. We added the following details (as you have seen above). To repeat:

 Data Collection and Management: Participants were recruited, and data was collected from November 2018 to August 2021. Data was collected on paper scales described above. The identifiers like names etc were not written on the questionnaires. These were entered in a different file and an ID was assigned to each participant. The data then was entered to iMann database. The database is password protected and only PIs and authorized research personnel can download or see the data. It is permanently stored at ICMR’s Bio-Informatics Division’s servers.

Results

Query: Table 1- community primary school n=41 please check n. if n=40 then overall N has to be reduced to 916.

Reply: Thank you for raising this query. Community primary school n is 41, the numbers in gender row are corrected and highlighted in Table 2 (now table 1 has been changed to table 2 as one more table as requested by Reviewer 1, has been added).

Discussion 

Query: Please bring in context of LMICs to the discussion part

Reply: Thank you for this very pertinent observation. We have now added text as follows:

Bauer et al. evaluated assessment tools for autism for their feasibility in LMICs and reported that most of these tools are not suitable for use in LMIC due to cost, copyright issues, restrictions on who can purchase and administer these tools and readability levels. In addition, these instruments do not have local norms. Furthermore, the assessments reviewed do not have local norms either. Only the M-CHAT-R/F can be used but it has narrow age range[40]. The IASQ is free to use and very short. It can be easily translated in the local language and adapted for use locally.

---

## [Editor Report · Decision Letter 1]

25 Sep 2023

Feasibility and acceptability of the Indian Autism Screening Questionnaire in clinical and community settings

PONE-D-22-30308R1

Dear Dr. Deshpande,

We’re pleased to inform you that your manuscript has been judged scientifically suitable for publication and will be formally accepted for publication once it meets all outstanding technical requirements.

Kind regards,

Engy Asem Ashaat

Academic Editor

PLOS ONE
---

## [Editor Report · Acceptance letter]

16 Nov 2023

PONE-D-22-30308R1 

Feasibility and acceptability of the Indian Autism Screening Questionnaire in clinical and community settings 

Dear Dr. Deshpande:

I'm pleased to inform you that your manuscript has been deemed suitable for publication in PLOS ONE. Congratulations! Your manuscript is now with our production department. 

Kind regards, 

on behalf of

Professor Engy Asem Ashaat 

Academic Editor

PLOS ONE